# Non-covalent control of spin-state in metal-organic complex by positioning on N-doped graphene

Bruno de la Torre[1,2], Martin Švec[1,2], Prokop Hapala[1], Jesus Redondo[1], Ondřej Krejčí[1], Rabindranath Lo[3], Debashree Manna[2,3], Amrit Sarmah[2,3], Dana Nachtigallová[2,3], Jiří Tuček[2], Piotr Błoński[2], Michal Otyepka [2], Radek Zbořil[2], Pavel Hobza[2,3] & Pavel Jelínek [1,2]

Nitrogen doping of graphene significantly affects its chemical properties, which is particularly important in molecular sensing and electrocatalysis applications. However, detailed insight into interaction between N-dopant and molecules at the atomic scale is currently lacking. Here we demonstrate control over the spin state of a single iron(II) phthalocyanine molecule by its positioning on N-doped graphene. The spin transition was driven by weak intermixing between orbitals with z-component of N-dopant ($p_z$ of N-dopant) and molecule ($d_{xz}$, $d_{yz}$, $d_z^2$) with subsequent reordering of the Fe d-orbitals. The transition was accompanied by an electron density redistribution within the molecule, sensed by atomic force microscopy with CO-functionalized tip. This demonstrates the unique capability of the high-resolution imaging technique to discriminate between different spin states of single molecules. Moreover, we present a method for triggering spin state transitions and tuning the electronic properties of molecules through weak non-covalent interaction with suitably functionalized graphene.

[1] Institute of Physics of the Czech Academy of Sciences, Cukrovarnická 10, 16200 Prague 6, Czech Republic. [2] Regional Centre of Advanced Technologies and Materials, Department of Physical Chemistry, Faculty of Science, Palacký University, Šlechtitelů 27, 78371 Olomouc, Czech Republic. [3] Institute of Organic Chemistry and Biochemistry of the Czech Academy of Sciences, Flemingovo nám. 2, 16610 Prague 6, Czech Republic. These authors contributed equally: Bruno de la Torre, Martin Švec, Rabindranath Lo.  Correspondence and requests for materials should be addressed to R.Z. (email: radek.zboril@upol.cz) or to P.H. (email: pavel.hobza@marge.uochb.cas.cz) or to P.J. (email: jelinekp@fzu.cz)

Graphene, a two-dimensional nanoallotrope of carbon, is currently at the forefront of scientific interest owing to its unique electronic, mechanical, optical, and transport properties[1,2]. It shows remarkable physical characteristics, such as large values of intrinsic mobility, Young's modulus, surface area, thermal and electric conductivity, and optical transmittance[1–6]. Hence, both pristine and chemically functionalized graphene have been found to be effective in a broad portfolio of applications, including electronics/optoelectronics, energy generation and storage, and various medical, chemical, catalytic, and environmental processes[7,8].

Doping of the graphene with foreign elements has been identified as an emerging strategy to open the zero band gap and tune its electronic, magnetic, and optical properties. Nitrogen in various structural configurations is the most frequently studied n-type dopant for improving the conductivity, transport features and magnetic properties of graphene, offering a multitude of applications in related fields like spintronics, energy generation/storage and nanoelectronics[9–13]. Importantly, the performance of graphene in these technologies is governed by the nitrogen concentration and its local environment in the graphene lattice (e.g., pyridinic, pyrrolic, graphitic), which determine its electronic features, including the closed or open electron shell arrangement.

In addition to allowing control over the intrinsic physicochemical properties of graphene, nitrogen doping can significantly affect its interaction with molecular species, as well as phenomena at the phase boundary of the graphene-molecule, which are particularly important in sensing and electrocatalysis applications[14–16]. Thus, graphene and its derivatives are attractive materials for sensing a vast variety of chemicals, (bio)molecules, and gases[17–19]. It is well accepted that adsorption of gas molecules onto the surface of graphene can be enhanced by nitrogen doping, causing changes in the local carrier concentration. Such graphene-based sensors can detect even a single gas molecule attached to or detached from the surface of graphene[20]. Similarly, N-doped carbon allotropes, including graphene, have been shown to act as superior electrocatalysts, establishing the emerging field of metal-free catalysis, with enormous prospects for fuel cells, metal-air batteries and oxygen-reduction reactions[21].

Moreover, very recent work using scanning probe microscopy (SPM) imaging techniques has demonstrated that N-doping of graphene can even tune the physicochemical properties of molecules adsorbed on its surface via non-covalent interactions. In particular, it was shown that 5,10,15,20-tetraphenyl-21H,23H-porphyrin molecules adsorbed on N-doped graphene undergo a downshift of their highest occupied molecular orbital (HOMO) and lowest unoccupied molecular orbital (LUMO) states if these molecules are located near to nitrogen defects[22]. This work indicated direct relevance of the graphene-molecule interaction for advanced organic electronics[22]. However, owing to the absence of a central metal atom, the possibility of discriminating or even controlling the spin state of a metal ion has remained challenging.

In the present work, we used high-resolution atomic force microscopy (AFM) with a non-magnetic CO-functionalized tip and inelastic spin excitation spectroscopy to characterize molecular electronic states. This state-of-the-art AFM technique has been shown to offer unprecedented spatial resolution, allowing investigation of the chemical structure of molecules[23–25] and atomic clusters[26], bond order analysis[27], molecular electrostatic field[28] or identification of different products of on-surface synthesis[29]. Here, we demonstrate that high-resolution images can also be used to discriminate different spin states of iron(II) phthalocyanine (FePc) molecules immobilized on N-doped graphene. The spin crossover was confirmed independently by inelastic spin excitation spectroscopy and theoretical quantum calculations. Importantly, we were able to control the spin state

by changing the molecular positioning on N-doped graphene. This is the first example, to the best of our knowledge, of a spin transition induced by non-covalent interaction of a molecule with doped graphene caused by reordering of iron $d$-orbitals without the use of an external stimulus, such as an external magnetic field, electric field, light, pressure, or temperature. Hence, we show the unique capability of doped graphene to behave as a non-covalent tuner of the electronic and spin properties of molecules.

## Results

**FePc adsorption on pristine and N-doped graphene**. We investigated the possibility of tuning the electronic and magnetic properties of iron(II) phthalocyanine molecules via non-covalent interaction with single nitrogen dopants in graphene. We used ion implantation followed by a thermal annealing procedure[30] to prepare N-doped graphene grown on a SiC(0001) surface with the majority of nitrogen defects in the substitutional positions. Hence, the term N-dopant refers to the substitutional (graphitic) variant in the following discussion, unless stated otherwise.

We found that the behavior of FePc was very specific depending on whether it was adsorbed on pristine or N-doped graphene. Submonolayer amounts of FePc deposited onto pristine graphene formed large and well-ordered flat islands, which were stabilized by intermolecular non-covalent interactions, as shown in the detailed scanning tunneling microscopy (STM) image in Fig. 1a. The molecules in the island were arranged in a square pattern with a lattice constant of about ≈1.40 nm, similar to FePc islands on HOPG[31] and G/Pt(111)[32]. On the other hand, deposition of FePc onto N-doped graphene resulted in strikingly different features. The large-scale STM image obtained at 5 K in Fig. 1b reveals single molecules and small clusters of molecules distributed over the substrate without any particular order. This strongly suggests that energetic barriers prevent lateral motion of FePc on N-doped graphene at low temperatures. Instead, they become pinned to dopant sites.

We performed lateral manipulation of individual molecules by SPM to identify their exact adsorption sites with respect to the location of the N-dopants. Figure 1c shows an STM image of a single FePc adsorbed on N-doped graphene acquired at –2.0 V. We intentionally moved the molecule laterally and then rescanned the same area at a lower bias of –0.05 V to allow resolution of the substitutional N-dopant that had been hidden below the molecule (Fig. 1d). The observed atomic contrast of the substitutional N-dopant agreed well with previous atomically resolved STM studies[30,33,34]. By comparing the pairs of images acquired before and after the FePc lateral movement, we could precisely determine the register of the FePc molecules with respect to the N-dopants beneath (marked by a red dot in Fig. 1c). We found a systematic preference for the FePc molecules to be located asymmetrically with respect to the N-dopant, as shown in the scheme in Fig. 1e.

The fact that we were able to easily manipulate molecules across the surface indicates that the FePc molecules have a low diffusion barrier across the surface. By revealing the N-dopants below the molecules, we learned that FePc was preferentially located in the vicinity of the N-dopants rather than on graphene. This observation suggests augmented interaction with the implanted N atoms in graphene, in agreement with ideas put forward from previous AFM spectroscopic measurements[30]. Manipulation of different molecular clusters revealed an N-dopant only under one of the molecules (Supplementary Figure 7). Therefore, the N-dopants seem to act as nucleation centers for FePc clusters on doped graphene. Concurrently, the random distribution of the N-dopants prevents any long-range ordering of the molecules.

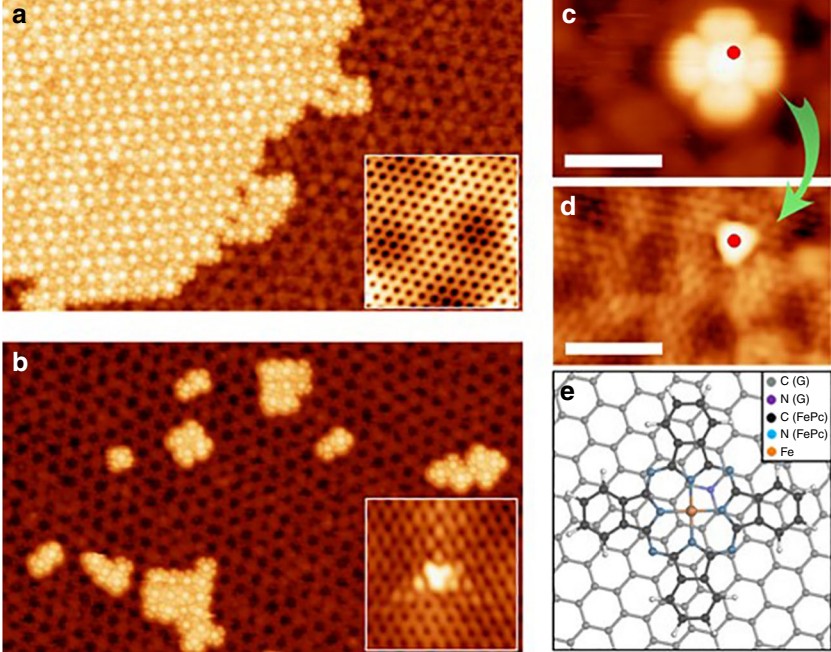

**Fig. 1** FePc adsorption on pristine and N-doped graphene at room temperature. 50 nm × 30 nm scanning tunneling microscopy (STM) images for FePc molecules on **a** pristine and **b** doped graphene ($V_b = -2.0$ V; $I_t = 10$ pA). The insets show high-resolution STM images of graphene and a N-dopant. **c** FePc molecule adsorbed on N-doped graphene before the manipulation event ($V_b = -2.0$ V, $I_t = 10$ pA). **d** The N-dopant revealed after controlled lateral manipulation of the FePc ($V_b = -0.05$ V, $I_t = 10$ pA). The red dot in **c** and **d** denotes the position of the N atom in the graphene lattice. The scale bars correspond to 2 nm. **e** Optimized model structure of FePc on N-doped graphene

**Effect of N-dopant on the FePc electronic structure**. The ability to control the position of single molecules by SPM also permitted investigation of the effect of the proximity of the N-dopant on their electronic structure. Figure 2a shows an example of the STM contrast change obtained after molecular manipulation, which depended on the distance from the N-dopant at which the FePc molecule was placed. In these experiments, we intentionally moved one of the molecules laterally to various positions with respect to the N-dopant, while the other remained stable on graphene and served thereby as a reference. The molecules were imaged after each step change in position at the same bias (–2 V) and the exact position of the N-dopant was resolved in the situation 1 by changing the bias voltage to −0.05 V below the plotted yellow dashed line, depicted on Fig. 2a. When the FePc was located over an N-dopant (situation 1 and 2), it exhibited a brighter contrast with four discernible lobes. In contrast, the FePc molecule on pristine graphene exhibited the characteristic eight lobes resolved at the π-rings of the molecule[35]. However, in situations 3 and 4, the influence of the N-dopant was diminished and contrast image of the manipulated FePc appeared similar to that of the reference unperturbed molecule. This indicates that the N-dopant had a very local effect on the electronic structure of FePc. Note that in situation 3, the center of the FePc was located only ≈1 nm from the N-dopant. However, the molecule showed almost identical appearance to that in situation 4, where FePc was located on pristine graphene.

The proximity of the N-dopant was also clearly manifested by changes in the differential conductance (d$I$/d$V$) spectra. Figure 2b shows d$I$/d$V$ spectra acquired from over the centers of two FePc molecules, one located on graphene and the other above an N-dopant. The FePc adsorbed on graphene exhibited two resonances, at –1.9 and 0.5 V, whereas the spectrum of the molecule adsorbed above the N-dopant showed shifts of the resonances to –1.2 and 0.3 V, effectively lowering the molecular gap by ~0.9 eV. This trend was very reproducible, as we observed it repeatedly

during more than ten sessions with different tips on several molecules (Supplementary Figure 6 and Supplementary Figure 5).

As mentioned above, the STM images of FePc molecules adsorbed at either graphene or N-dopants and acquired at energies near the d$I$/d$V$ resonances, showed a different contrast and number of lobes (Fig. 2c). These types of images are typically attributed to single particle HOMO and LUMO signatures in the d$I$/d$V$[36]. However, according to recent studies on similar molecular systems[37,38], these resonances should not be directly interpreted as such. Instead, they reflect the many body spectral functions of electron tunneling processes in and out of the molecule. Nevertheless, the differences in molecular shape measured at the resonance indicate substantial modification of the electronic structure of FePc molecules near the N-dopant compared to those located far away.

**Local electric field of the N-dopant and its effect on the FePc molecule**. One explanation for the altered electronic structure could be varying charge transfer between the molecule and substrate depending upon the adsorption site[37]. However, this hypothesis was discarded because of the location of the d$I$/d$V$ resonances far from the Fermi level (Fig. 2b) and by our Kelvin probe force microscopy (KPFM) measurements. Figure 3e presents the frequency shift dependence on the applied bias acquired at a constant height over FePc molecules adsorbed on pristine graphene and above an N-dopant. Over the FePc molecules, we did not observe any substantial variation of the local contact potential difference (LCPD). At the same time, the reference spectra obtained above graphene and the N-dopant showed a significant difference due to the doping effect of the N atom, as seen in the LCPD map (Fig. 3a, b). In particular, the LCPD over the N-dopant shifted to a lower value, reflecting a lower local work function induced by the positive net charge localized at the dopant.

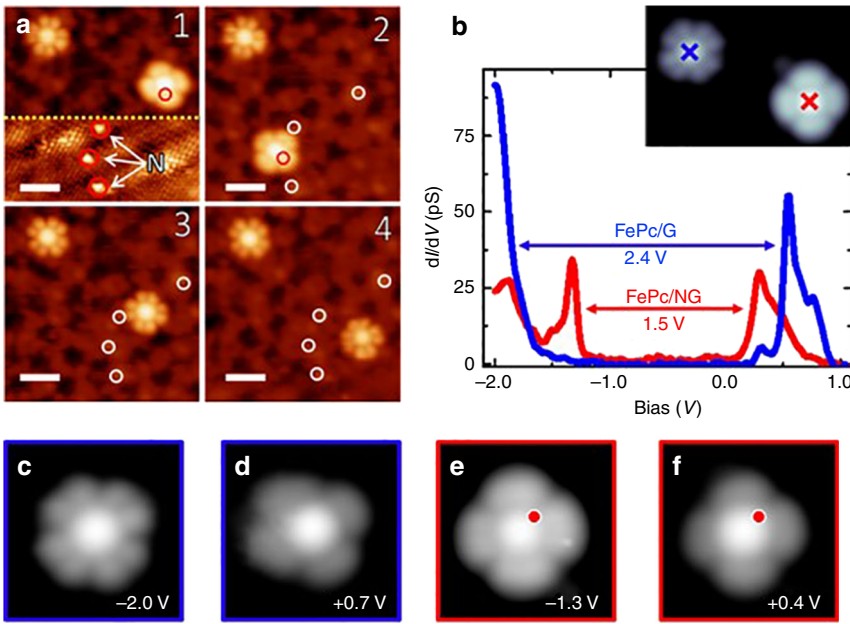

**Fig. 2** Effect of N-dopant on the FePc electronic structure. **a** STM images acquired after steps of controlled lateral manipulation of FePc on N-doped graphene. From the initial situation (1), where a FePc molecule is adsorbed on an N-dopant, the molecule was moved to a different configuration on another N-dopant (2), then to a distance of 1.3 nm from any N-dopant (3), and finally to pristine graphene (4). The circles mark the positions of the N-dopants. The FePc molecule in the top left is located on pristine graphene and served as a reference. The scale bar corresponds to 2 nm. **b** d$I$/d$V$ spectra acquired at the center of FePc on pristine graphene (blue) and on an N-dopant (red). 3.6 nm × 3.6 nm STM bias-dependent images ($I_t$ = 10 pA) of FePc on pristine graphene (**c**, **d**) and on an N-dopant (**e**, **f**), acquired at biases near the resonances shown in the d$I$/d$V$ spectra (**b**)

Our experimental observations were supported by both periodic and cluster DFT calculations (see Supplementary Methods and Supplementary Figure 1 for more details). FePc molecules adsorbed non-covalently on pure and N-doped graphene in very similar planar configurations, located about 3.3 Å above the surface. According to our calculations, the interaction energy of FePc was 7 kcal/mol higher on the N-dopant than on pristine graphene (see Supplementary Table 1). Notably, there was an energy barrier of 8 kcal/mol preventing lateral motion of FePc near the N-dopant, whereas it was negligible on pristine graphene. This explains the FePc stabilization in the vicinity of the dopants observed in the experiment. The optimized position of FePc showed that the central iron atom was laterally displaced by ~2 Å from the N-dopant (Fig. 1e) with the whole FePc molecule tilted slightly with respect to the surface, in perfect agreement with our experimental observations (Supplementary Figure 4).

The non-centric position of the FePc molecule with respect to the N-dopant was caused by interaction of its positive charge with the $Fe^{2+}$ ion and the negatively charged pyrrolic-N in the FePc molecule. The presence of the positive charge on the N-dopant was evident in both our KPFM measurements (discussed above) and the electrostatic Hartree potential obtained from DFT calculations. Indeed, the electrostatic potential over the N-dopant obtained from DFT calculations (Fig. 3d) showed similar character to the LCPD map (Fig. 3b). The positive charge on the N-dopant originated from incorporation of one of its valence electrons into the linear graphene π-band, causing its partial delocalization and subsequent shift of the Dirac cone below the Fermi level[39].

**Spin crossover of FePc positioned on an N-dopant**. To gain more insight into the influence of the N-dopant on FePc molecules, we performed high-resolution AFM imaging with a CO-functionalized tip[23]. Figure 4a shows an AFM high-resolution

image of three FePc molecules, two of them (right and left side of image) were adsorbed on pure graphene, whereas the other (middle) molecule was adsorbed on an N-dopant (the red dot indicates the exact location of the N-dopant, determined by removing the molecules after the AFM measurements). For all three molecules, the four peripheral benzenes were resolved similarly with almost equal brightness, which indicates that the molecules were adsorbed in a nearly planar configuration[40] and at very similar distances above the surface (Supplementary Figure 9). Note, one benzene of the FePc molecule at the N-dopant appeared slightly brighter, suggesting a very small tilt of the whole molecule with respect to the substrate. This observation is in good agreement with the optimized structures obtained from the total energy DFT calculation discussed above.

Interestingly, the cores of the molecules displayed more significant variation. A cross-shaped feature was observed in the middle of both molecules adsorbed on pure graphene (Fig. 4b), whereas for FePc adsorbed on the N-dopant, a square-like feature was clearly resolved (Fig. 4c). This variation in the AFM contrast was very reproducible with different CO-functionalized tips and molecules (Supplementary Figure 8). To understand the origin of this contrast difference, we have to consider the mechanism of the high-resolution imaging[23,41,42]. AFM images reflect the distribution of the total electron density within the inspected molecule[43] and its variation can significantly affect the submolecular contrast[28]. Our total energy DFT calculations showed a tiny shortening of internal Fe–N bond by ~1% when located on the N-dopant accompanying the transition between the triplet and singlet states. The whole molecule is slightly tilted, due to asymmetric position of the N-dopant with respect to center of the molecule. However, importantly, we do not observe any vertical relaxation of internal Fe or N atoms out of the molecular plane (Supplementary Figure 4). Therefore, we can rule out that the contrast variation is caused by the internal vertical relaxation of Fe and N atoms out of molecular plane.

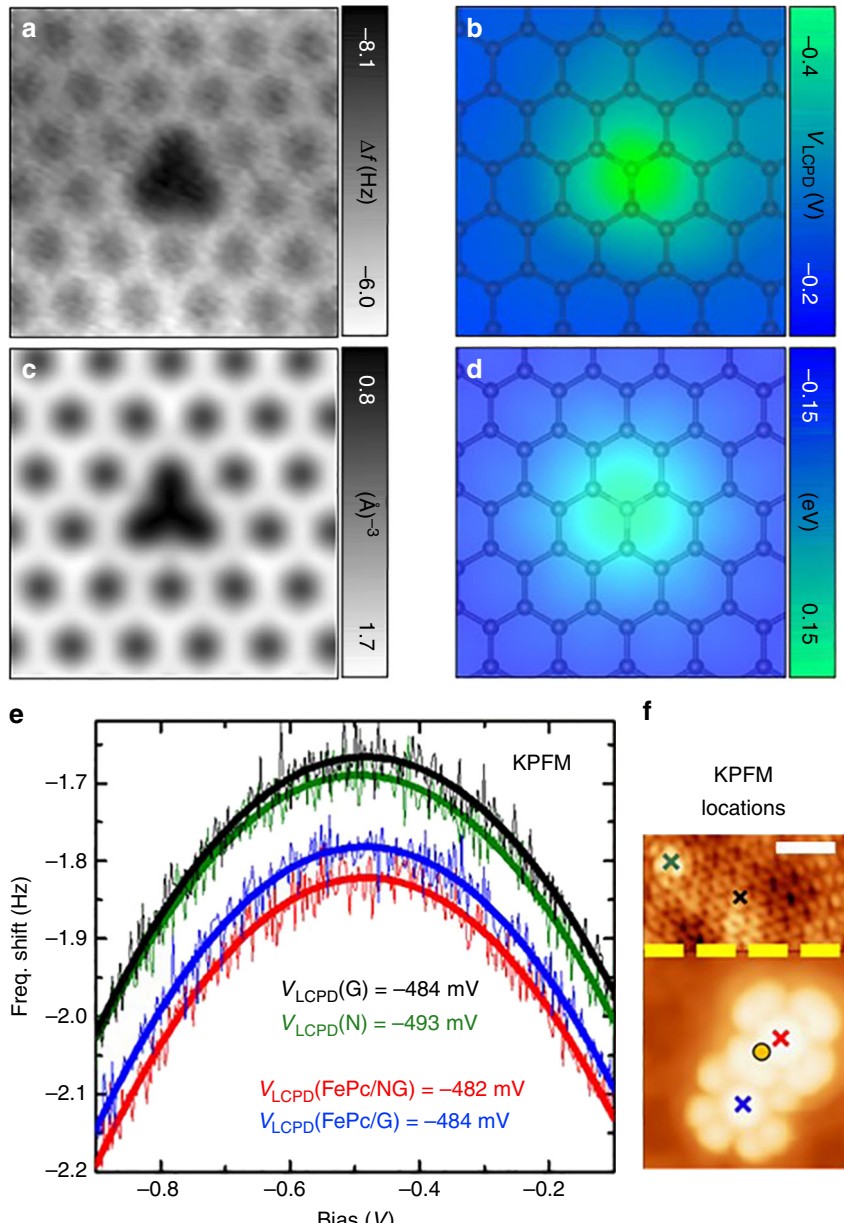

**Fig. 3** Charge distribution in the N-dopant. **a** Atomically resolved constant-height AFM image of a single N-dopant in graphene. **b** Local contact potential difference ($V_{LCPD}$) map acquired at a constant height immediately after (A) over the same area but with the tip retracted by 200 pm. **c** Calculated total electron density of N-doped graphene. **d** Calculated Hartree potential of a single N-dopant in graphene. **e** Frequency shift dependence with bias voltage acquired at the center of a FePc molecule adsorbed on pristine graphene (blue) and at an N-dopant (red), as well as on an N-dopant (green) and on a C atom in pristine graphene (black). The four measurements were acquired at the same tip-sample distance and errors in $V_{LCPD}$ fittings are about 1.5 mV. **f** STM image with the measurement locations, recorded at $V_b = -0.05\,V$ (top part) and $-2.0\,V$ (bottom part). The orange circle marks the position of the N-dopant beneath the FePc molecule determined from the LCPD measurements. The scale bar corresponds to 1 nm. The data plotted in (**b**) and (**e**) were acquired in different sessions with metallic tips at different tip-sample separations

Consequently, we focused our attention on the electronic structure of the FePc molecule, which is well known to change, depending on its environment and external stimuli[44–50]. The results from d$I$/d$V$ spectroscopy provided strong evidence that the electronic structure of FePc was modified in the proximity of an N-dopant, which may be accompanied by variation of the molecular spin state. To investigate such a possibility, we compared total electron densities of free standing FePc molecule in triplet and singlet states calculated using both DFT and the multi-configurational self-consistent field method (MCSCF). Figure 4d and e show that there was substantial density variation in the center of the molecule. In the singlet state, there was a substantially lower density in the central region, contrary to the triplet state, which showed an excess of electrons. This difference originated from differing occupancy of the $d_z^2$ Fe orbital, as shown in Fig. 4f. Therefore, one may expect that when using a CO-functionalized tip, the relaxation alters due to changes in the Pauli and electrostatic interactions at close distances to the molecule[42]. Indeed, simulated AFM images with the electron density of the triplet and singlet states of FePc reproduced the experimental images obtained for molecules on graphene and the N-dopant, respectively (Fig. 4d, e) (see Supplementary Table 2). This is a remarkable result since it extends already outstanding capabilities of high-resolution AFM technique by a possibility to

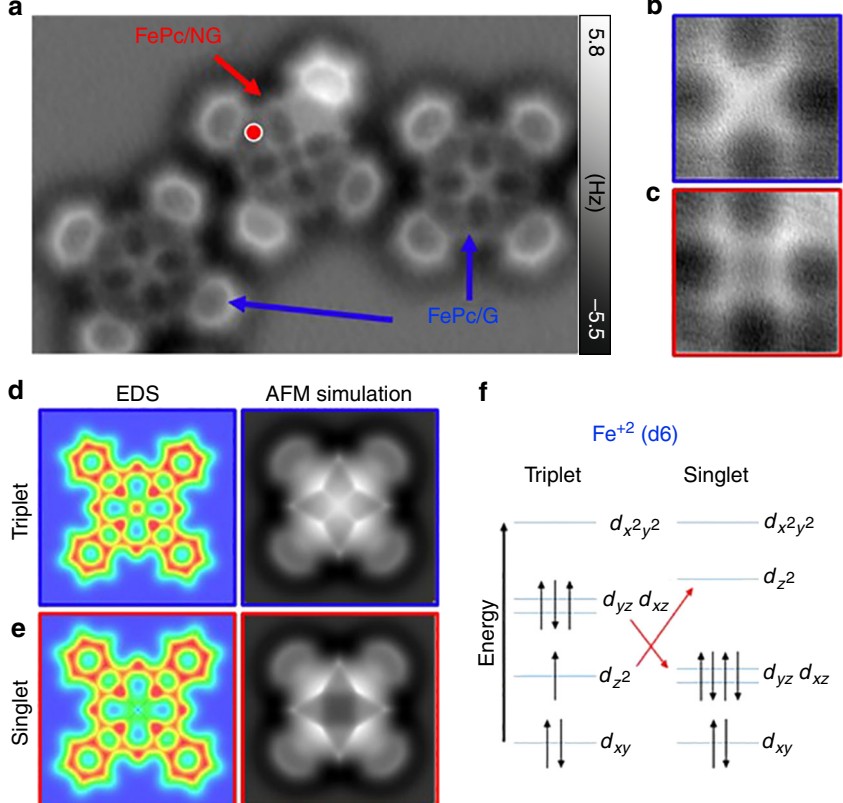

**Fig. 4** Spin crossover on a FePc molecule. **a** Constant-height AFM image of three FePc molecules located on pure graphene (blue arrows) and above an N-dopant (red arrow). The red dot marks the exact position of the N-dopant. **b** and **c** AFM images of the cross and square features at the core of a FePc molecule on pristine and N-doped graphene, respectively. **d** and **e** Calculated electron density and simulated AFM image of FePc when the Fe atom is in the triplet and singlet spin state, respectively. In the triplet state, a cross-shaped feature was observed at the core of FePc, whereas a square feature was obtained in the singlet state due to electron depletion at the center of the molecule. **f** Occupation of $Fe^{+2}$ orbitals in the triplet and singlet states. (Note: the other triplet state characterized by the iron $d$-orbital occupation $d_{xy}^2 d_{z^2}^2 d_{xz}^1 d_{yz}^1 d_{x^2-y^2}^0$ provides the same scenario). Since, $d_{z^2}$, $d_{xz}$, and $d_{yz}$ orbitals are close in energy, the $d_{xy}^2 d_{z^2}^2 d_{xz}^1 d_{yz}^1 d_{x^2-y^2}^0$ and $d_{xy}^2 d_{z^2}^1 (d_{xz} d_{yz})^3 d_{x^2-y^2}^0$ states are almost isoenergetic. The proximity of the N-dopant affects the atomic orbital distribution

discriminate different molecular spin states without need for magnetic tips. Although the case of FePc may seem very specific, we want to stress that a similar approach should be possible whenever changes in the spin state are associated with a considerable local change in electron density within the molecule. However, the unambiguous confirmation of the spin crossover should be always confirmed by independent measurements, such as inelastic spin excitation spectroscopy.

To reinforce our interpretation of the spin transition, we carried out inelastic spin excitation spectroscopy[51]. We acquired d$I$/d$V$ spectroscopy of FePc molecules adsorbed on pristine graphene and on an N-dopant near the Fermi level, see Fig. 5. Two molecules show the distinct characteristic STM contrasts, very similar to those presented in Fig. 2, indicating the different electronic configuration. The d$I$/d$V$ spectroscopy (see Fig. 5b) acquired for these two molecules with the same tip termination reveals very distinct character. In the case of the FePc molecule located on the N-dopant, there is no evidence of any symmetric steps in bias voltages associated with the spin excitation signal. This observation fully supports our claims that the molecule is found in singlet state, where the spin excitations are not expected. On the other hand, in the case of the FePc on pristine graphene it is found in the triplet state. The degeneracy of the triplet state is partially lifted by spin–orbit interaction which gives rise to the lowest spin state $S = 1$, $m_s = 0$ and doubly degenerated $S = 1$, $m_s = \pm1$ states. This splitting occurs even in absence of magnetic

field and it is called zero-field splitting.[43] This effect is well described by interaction Hamiltonian $\mathcal{H}_{SO} = DS_z^2 + E\left(S_x^2 - S_y^2\right)$, where $D$ is vertical magnetic anisotropy (origin of the zero-field splitting) and $E$ is in-plane magnetic anisotropy, which for planar molecules typically vanishes. Consequently, the Hamiltonian reduces to $\mathcal{H}_{SO} = DS_z^2$ and inelastic spin excitation spectra of FePc molecule in the triplet $S = 1$ state should show a single inelastic signal symmetric in bias voltage of the magnitude $D$. Indeed, we observe a symmetric step-wise increase of the conductance at bias voltages approximately $\pm7$ meV. Consequently, we interpret this feature as inelastic spin excitations of the $S = 1$ spin multiplet ($m_s = 0$ and $m_s = \pm1$) due to the zero field splitting driven spin–orbit interaction[44]. We would like to stress, that very similar spectra were also obtained for Fe-tetraphenyl-porphyrin molecule[52], which were also attributed to inelastic spin excitation the $S = 1$ spin state. Therefore, we believe that the additional inelastic spin excitation measurements fully support the scenario of the spin transition of FePc from triplet (pristine graphene) to singlet (on N-dopant) state.

The presence of the spin transition was also supported by our DFT calculations. The most stable electronic configuration of FePc adsorbed on pure graphene corresponded to the high-spin (triplet) state. Strikingly, in the proximity of the N-dopant, the low-spin (singlet) state was found to be about 5 kcal/mol more stable than the triplet state. We attributed this effect to

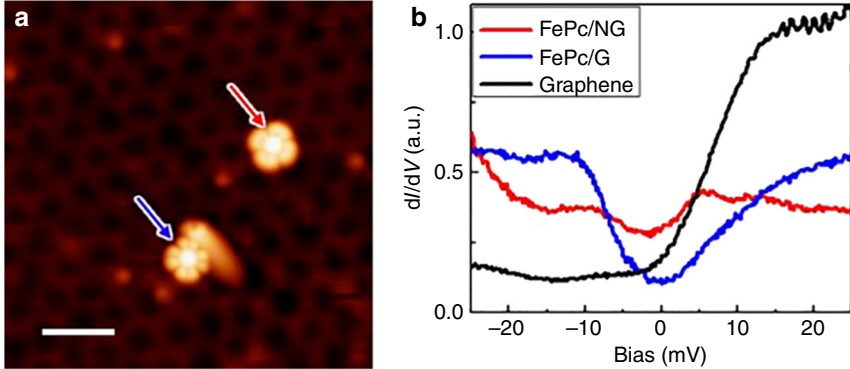

**Fig. 5** Inelastic spin excitation signal on FePc molecules. **a** STM topography image of FePc molecules on pristine graphene (blue) and on an N-dopant (red). The FePc molecule on pristine graphene is immobilized near a graphene bubble. Image parameters: $V_b = -2.0$ V; $I_t = 10$ pA. **b** d$I$/d$V$ spectroscopy near the Fermi level on both FePc molecules adsorbed on pristine graphene (blue) and on graphitic N-dopant (red) and on pristine graphene (black) substrate for comparison. Spectroscopies were acquired at the same tunnel resistance (500 MΩ) with voltage modulation of 0.5 mV at 170 Hz

weak intermixing between orbitals with $z$-component, i.e., $p_z$ of the N-dopant points outward from the surface and the ($d_{yz}$, $d_{xz}$) and $d_z^2$-like molecular orbitals of Fe. This causes an upward energy shift of the $d_z^2$-orbital with respect to ($d_{yz}$, $d_{xz}$), which promotes electron transfer from $d_z^2$ to ($d_{yz}$, $d_{xz}$), leaving the $d_{z2}$ orbital empty with a closed shell electron configuration, as shown in Fig. 4f. This orbital reordering of the $z$-component $d$-orbitals is facilitated by the fact they are very close in energies (Supplementary Figure 2). A relatively large distance between the Fe atom and the N-dopant (around 3.7 Å) prevents significant overlap between the orbitals, and thus formation of a strong dative bond. In our case, there was only a weak interaction between Fe and the N-dopant (Wiberg bond index of 0.01 compared with 0.57 for dative bonds between Fe and N within FePc). On the other hand, analysis of electronic structure of pristine and N-doped graphene reveals, in the case of graphitic N-dopant, a presence of a localized state near the Fermi level (see Supplementary Figure 3c)). The charge density of this localized state substantially protrudes out of the surface compared with the pristine graphene (see Supplementary Figure 3a and b). Therefore, this charge density localization enhances the interaction between the protruding $p_z$-orbitals of the graphitic N-dopant and three adjacent C atoms and $z$-component $d$-orbitals of Fe contrary to pristine graphene. This effect is also supported by lowering of calculated Wiberg bond index between FePc and pristine graphene by one half with respect to the case of FePc on N-defect. Consequently, we attribute the orbital reordering and the related spin transition to this effect of charge localization.

## Discussion

The possibility that the spin crossover was due to the local electrostatic field of the N-dopant was ruled out by analysis of a double substitutional nitrogen defect in the *para* configuration, which has an even stronger local electrostatic field. Supplementary Figure 10 shows a series of STM pictures where the FePc molecule is manipulated between a single and *para* N-defect. When the molecule was located near the *para* N-defect, it exhibited a similar STM contrast and d$I$/d$V$ spectrum to those of molecules located on pristine graphene. However, the contrast and d$I$/d$V$ spectrum significantly changed when the molecule was manipulated onto a single N-defect due to the spin crossover. Again, this scenario was supported by our DFT calculations, which predicted that the molecule would be in the triplet state when located over the *para* N-defect. These results also strongly support the role of an open-shell like character of the single

substitutional N-defect. The *para* N-defect has a closed-shell electronic structure which is more stable than the biradical one. The unique chemical activity of graphitic N-defects for spin crossover was further supported by theoretical analysis with different model cases, including a closed-shell pyridinic defect (for details, see Supplementary Materials), for which proximity to the molecule again stabilized FePc in the triplet state.

These results demonstrate the diverse chemical activity of nitrogen impurities incorporated into graphene, opening a new way for not only tuning the electronic properties of molecules but also chemical functionalization of graphene. The unique character of the substitutional N-defect allows not only stabilization of the FePc molecule in its vicinity but is also responsible for the spin crossover of FePc. For comparison, such spin crossover of FePc is usually triggered by strong ligands or ligands with unpaired electrons forming covalent dative bonds[53,54] in classical coordination chemistry. In our case, we demonstrated that such a transition can be achieved just by sliding the molecule over the surface of N-doped graphene. Thus, the electronic states of FePc molecules on graphene can be tuned locally by means of weak non-covalent interaction with N-dopants causing reordering of the selected iron $d$-orbitals driven by their weak hybridization with the $p_z$ orbital of a single graphitic nitrogen defect. This offers a way for controlling the spin state of a molecular system by simple positioning of the molecule onto a suitably functionalized graphene substrate. We also showed that high-resolution AFM using a non-magnetic CO-functionalized tip can distinguish between different spin states of molecules on a surface. This extends further the outstanding capabilities of the technique and opens new possibilities for studying magnetic properties at the single molecule level.

## Methods

**STM/AFM measurements.** STM/AFM measurements were carried out in a UHV chamber equipped with a low-temperature STM/AFM with qPlus tuning fork sensor operated at 5 K (Createc GmbH). During the AFM measurements, a Pt/Ir tip mounted onto the sensor (frequency ≈30 kHz; stiffness ≈1800 N/m) was oscillated with a constant amplitude of 50 pm. To obtain high-resolution AFM/STM images, prior to the experiment, the tip was functionalized with a CO molecule gathered from a Au(111) surface. d$I$/d$V$ measurements were acquired with the conventional lock-in technique with an amplitude modulation of 20 mV at 960 Hz. The metallic tip for d$I$/d$V$ spectroscopy was formed by repeated indentations into a Au(111) surface until the surface state was clearly visible. Inelastic spin excitation spectroscopies were acquired at 500 MΩ of tunnel resistance with voltage modulation of 0.5 mV at 170 Hz. A graphene sample was prepared on a SiC (0001) substrate by annealing in Ar and was degassed in a UHV system at ≈800 °C[55]. N atom implantation was achieved by sputtering the graphene sample with N atoms accelerated at 120 eV and subsequent annealing to ≈800 °C. Iron(II)

phthalocyanine (FePc) was thermally evaporated from a Ta pocket in UHV onto the graphene surface, which was kept at RT. After deposition, the sample was briefly annealed at temperatures below 200 °C to increase the surface mobility. STM/AFM images were analyzed using WSxM and WSPA software[56].

**DFT calculations**. Optimized structures of FePc molecules on N-doped and pristine graphene were calculated using both the cluster and slab models in the Turbomole, Gaussian, and VASP program packages based on ab initio density functional theory (DFT) (for details, see Supplementary Methods).

**AFM simulations**. Theoretical AFM images were calculated using a probe particle AFM simulator . Further details about the Methods can be found in the Supplementary Methods.

**Data availability**. The data that support the findings of this study are available from the corresponding authors on request.

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

## Acknowledgements

This work was part of the Research Project RVO: 61388963 of the Institute of Organic Chemistry and Biochemistry, Czech Academy of Sciences. The authors also acknowledge support of the Czech Science Foundation under projects no. 16–16959S and 17-24210Y, and the Ministry of Education, Youth and Sports of the Czech Republic under projects LO1305, LM2015087, LM2015073, and CZ.02.1.01/0.0/0.0/16_019/0000754. P.J. acknowledges support from the Czech Academy of Sciences through a Praemium Academiae award. M.O. acknowledges to ERC consolidator grant 683024 from the European Union's Horizon 2020 Research and Innovation Programme.

## Author contributions

B.T. and M.Š. performed SPM experimental measurements. J.R. prepared graphene samples. Pr.H., O.K., and P.J. preformed AFM simulations and their analysis. R.L., D.M., A.S., D.N., P.B., M.O., P.J., and Pa.H. performed DFT simulations. D.M., D.N., and Pa.H performed MCSCF calculations. B.T., R.Z., Pa.H. and P.J. conceived and designed the experiments. J.T. participated in manuscript writing. B.T., R.Z., Pa.H., and P.J. wrote manuscript. All authors discussed the results and commented on the manuscript.

## Additional information

**Competing interests:** The authors declare no competing interests.

