## [Peer Review File · Nature Communications]

Reviewers' comments:

Reviewer #1 (Remarks to the Author):

The manuscript reports on the adsorption of FePc molecules on nitrogen doped graphene. The properties of FePc adsorbed near single graphitic N-dopants are compared with FePc on pristine areas using a combination of local probe methods, i.e., scanning tunneling microscopy (STM) and spectroscopy, atomic force microscopy (AFM) as well as Kelvin probe force microscopy. Near the dopant sites the molecules exhibit pronounced changes in their local electronic densities that are visualized with STM and AFM. The authors claim that these changes are due to distinct spin states of the Fe centers, which are induced by the weak interactions of the adsorbates with the N-dopant sites and support their conclusions with detailed ab-initio calculations for a model system. Hence, the major claims of the manuscript are the control of the spins state of adsorbates by the dopants in graphene and the imaging of the change of spin state using (non-spin-polarized) AFM.

The manuscript is clearly written and the presented data and calculations are of high quality. The findings are generally very interesting and of high relevance for graphene-based devices and control of surface chemistry in general. The experimental data provides clear evidence for the change of electronic character of the molecules and the manipulation experiments with the STM tip are convincing. To strengthen the observations the authors should provide a number on how many different molecules were investigated this way.

The theoretical modelling provides a detailed analysis of the observations and appears to describe the experiments well assuming a change of spin state. However, to claim that one can experimentally discriminate different spin states of the adsorbates one requires an independent and established method, e.g., spin-polarized STM, to prove that the distinct spin states indeed exists. As it is now, the presented theory is in line with the observations but this is no experimental proof, also given the adaption made in the idealized model. So far, only changes in the electronic states are observed.

As a remark: In spin transition molecules, orbital and spin degrees of freedom are strongly intertwined and hence a spin state change is always accompanied with an electronic change that can be observed with standard SPM techniques. Thus, the proposed system may not be such a good candidate to demonstrate the discrimination of spin states by non-spin-polarized SPM techniques in general.

However for the presented system, I suggest to provide independent experimental evidence of the spin state of the FePc molecules before publication of the manuscript can be granted.

Reviewer #2 (Remarks to the Author):

The authors present here a combined experimental and theoretical work on control of spin state in a FePc molecule deposited on nitrogen doped graphene. In particular, they show that when the molecule is positioned on top of a nitrogen atom, it experiences a spin transition of the central Fe atom, from the triplet to the singlet state. This particular behavior has been probed using a CO decorated AFM tip, which reveals different intramolecular features with respect to the spin state of the molecule, and therefore with respect to the positioning of the molecule on top of a nitrogen atom or not. Moreover, these experimental findings are supported by DFT calculations, exhibiting the non-covalent character of the molecule-surface or molecule-nitrogen interaction.

I find this article very interesting and very innovative, since, as claimed by the author, the spin state of the molecule can be controlled by the sliding of this latter on the graphene surface using a standard STM tip. Also, the probe of the spin transition inside the molecule is performed in a very elegant way using the CO decorated AFM tip that shows the spin transition through different features inside the molecule. Consequently, I think this article deserves to be published in Nature Communication. I have however some questions or comments in order to understand well the

Physics of this system as well as to improve the manuscript.

To my opinion, the fundamental point that has to be explained in more details is the physical origin of the spin transition when positioning the molecule on top of a nitrogen atom. Indeed, in their abstract, the authors talk about “hybridization of the d-orbitals of iron and the pz orbital of a single graphitic nitrogen” and then on page 15 when explaining the obtained result, they talk about “weak intermixing between orbitals with z-component ... of the N-dopant ... and ... Fe”. These two assumptions seem to be rather contradictory, since in the first case one would think about a covalent bonding, whereas in the second, one would think more in an electrostatic effect. At least one of the two sentences has to be corrected. But more importantly, I would request the authors to define more this interaction. Is this due to van der Waals forces between the molecule and the nitrogen atom? Is it only an electrostatic effect at the van der Waals distance between the molecule and the nitrogen atom? How do the authors define the concept of “weak intermixing between orbitals”, and besides, what would be a “strong intermixing”? Also, what is the interaction responsible for the promotion of an electron from the orbital dz² to the orbitals (dyz, dxz)? In that respect, I am also a bit confused with the spin repartition in the d orbitals of the triplet state. Indeed, naively one would expect an extra spin down in the dz² orbital letting two spins up in the (dyz,dxz) orbitals, gaining energy. Can the authors comment about it? This peculiar repartition could be related to some molecular distortion, and a change in the distortion due to the interaction with nitrogen could also be responsible for the spin transition by increasing the level splitting. Therefore, I would request the authors to accurately analyze the calculated geometries of the molecule on top of nitrogen and on top of pristine graphene, in order to observe some bond lengths differences or distortions. By the way, the authors talk about a possible tilt of the FePc molecule on the N-dopant, this tilt could also be the expression of this geometry change due to the interaction with the N dopant.

I think also that a representation of the calculated isoelectronic density of states between the graphitic nitrogen and the iron atom, compared to the case of standard graphene and the iron atom is very important to better understand this concept and more particularly, the non-significant overlap. Actually, it seems believable that at 3.7 angströms distance the interaction is so weak that there is no significant overlap, but on the other hand, it is more complicated to understand the important energy shift of the dz² orbital and the consequent electron transfer leading to the spin transition. Consequently, such a representation might help the reader to understand this process.

I also have some minor comments. Regarding the methodology, what is the interest in performing cluster with a circumcircumcoronene molecule and periodic calculations instead of using directly the graphene unit cell and periodicity? In particular, the authors have used cluster calculations to prove that there is no significant charge transfer, but does the charge transfer between the molecule and the circumcircumcoronene reproduce well the charge transfer between the molecule and graphene?

Finally, there are several typos in the text that need to be corrected, the legend of Figure 3 (E) has to be corrected (confusion between red and green curves), and the energy units should probably be set in eV/molecule.

Reviewer #3 (Remarks to the Author):

In the manuscript “Non-Covalent Control of Spin-State in Metal-Organic Complex by Positioning on N-Doped Graphene and Its SPM Discrimination”, the authors Bruno de la Torre, et al. study the influence of the presence of N-dopants in graphene on the adsorbed iron(II)phthalocyanine (Fe-Pc) molecule.

They use a non-contact atomic force microscope with 5 K base temperature to measure the local contact potential difference with Kelvin probe between Fe-Pc molecules adsorbed on top of a N-defect or on top of pristine graphene. Furthermore, by functionalizing the tip apex with a CO molecule they detect between molecules adsorbed on these two sites saddle differences in the frequency shift channel of constant height images. Using DFT calculations they claim that the difference is due to a change of the spin state of the magnetic molecule.

Unfortunately, the authors fail to prove their claim. From the experimental point of view I am surprised that the authors haven't performed dI/dV measurements around the Fermi-edge with higher resolution as the ones shown in figure 2b. At small voltages the spin states of Fe-Pc can be revealed either via spin-excitations as seen p.ex. in ref. 43 for Fe-Pc on Cu(110)-(2x1)-O or by the presence of the Kondo state as seen p.ex. in ref. 44 for Fe-Pc on Au(111). Such a measurement would enable to unambiguously prove their claim.

Here I would like to mention that many experiments have seen 2 or more types of Fe-Pc molecules depending on the precise adsorption configuration on different surfaces. For example Tsukahara et al. (ref. 43) found to different adsorption sites on Cu(110)-O which changed the magnetic anisotropy of the molecule but NOT the spin state. Gao et al. (PRL99,106402(2007)) was the first who observed site dependent Kondo features for Fe-Pc on Au(111). Subsequently, Ben Warner et al. (Nat.Nano 10, 259 (2015)) showed that on CuN/Cu(110) the molecule can exhibit strong negative-differential conductance with clear magnetic origin.

To summarize, only observing two different topographic appearances is not sufficient for the claim of changing the spin state. Therefore, in the present form I can not recommend the paper for publication. However, if the authors provide additional evidence then the situation might change.

Apart from the main critique I would like to authors prior resubmission to clarify the following point.

- Figure 2a shows the manipulation of Fe-Pc. Why are the N-defects which are clearly visible in (1) no longer visible in (2)-(4)? What is the dashed line indicating?

- Figure 3 shows KPFM measurements. Why does the map in (b) has a dynamic range of $\sim 0.2V$ but the $df(V)$ measurements on the dopant and pristine graphene varies only by $\sim 9mV$? What are the errors of the parabolic fit? Is this small difference outside the uncertainties? Panel (e) also shows an STM image of two molecules on which the $df(V)$ spectra have been obtained. Why did the authors chose two molecules which clearly overlap?

- In lines 255-257 the authors write: "This is a remarkable result since it demonstrates the capability of the high-resolution AFM technique to distinguish between different molecular spin states without the need for a spin-polarized tip." This sentence is not correct. It is not necessary to have a spin-polarized tip for distinguishing between different spin states. It is broadly used state-of-the-art to determine the spin state with dI/dV measurements.

Referee 1 comment:

-The manuscript is clearly written and the presented data and calculations are of high quality. The findings are generally very interesting and of high relevance for graphene-based devices and control of surface chemistry in general. The experimental data provides clear evidence for the change of electronic character of the molecules and the manipulation experiments with the STM tip are convincing. To strengthen the observations the authors should provide a number on how many different molecules were investigated this way.

Authors reply:

We thank the referee for the positive response. We observed the variation of the electronic structure of FePc by positioning on doped graphene during different session and with different tips. In total, we investigated more than 10 systems, which showed very similar behavior. In particular, Figure S5 shows 4 different sets of dI/dV spectroscopies acquired during different experimental sessions. Each set shows two dI/dV spectroscopies acquired on two different molecules, one adsorbed on a N-dopant while the second is positioned on pristine graphene. The data reveal high-reproducibility of the electronic structure variation upon positioning of FePc molecule near the substitutional N-dopant. The same holds for the high-resolution imaging with CO-functionalized tip. Fig S8 represents different AFM images acquired during different sessions, where the variation of the submolecular contrast in the central part is clearly visible.

Action:

We have added the following sentence in the Main text in the paragraph where dI/dV spectroscopy is mentioned: “This trend was very reproducible, as we observed it repeatedly during various sessions with different tips **on several molecules, as shown in Fig. S6.**”

Referee 1 comment:

- The theoretical modelling provides a detailed analysis of the observations and appears to describe the experiments well assuming a change of spin state. However, to claim that one can experimentally discriminate different spin states of the adsorbates one requires an independent and established method, e.g., spin-polarized STM, to prove that the distinct spin states indeed exists. As it is now, the presented theory is in line with the observations but this is no experimental proof, also given the adaption made in the idealized model. So far, only changes in the electronic states are observed.

As a remark: In spin transition molecules, orbital and spin degrees of freedom are strongly intertwined and hence a spin state change is always accompanied with an electronic change that can be observed with standard SPM techniques. Thus, the proposed system may not be such a good candidate to demonstrate the discrimination of spin states by non-spin-polarized SPM techniques in general.

However for the presented system, I suggest to provide independent experimental evidence of the spin state of the FePc molecules before publication of the manuscript can be granted.

Authors reply:

We agree with the referee that an independent experimental evidence should prove the spin state of the FePc molecules located on pristine graphene and on a nitrogen dopant. We acknowledge the relevance of this request since referee #3 also asks for an extra experimental evidence of the spin transition.

One possibility would be to perform XMCD measurements. Unfortunately, a low concentration of the N-dopants ($> 1\%$) would not allow to acquire sufficiently strong signal to provide reliable experimental evidence of the spin transition of FePc molecules.

Therefore, we decided to perform additional experiments based on STM inelastic spin excitations of the FePc molecules following a suggestion of the Referee #3, which can provide the direct evidence of the spin transition. We acquired dI/dV spectroscopy of FePc molecules, see in Fig. R1, adsorbed on pristine graphene (blue arrow) and on a N-dopant (red arrow) near the Fermi level. The corresponding spectra in Fig. R1 b were obtained with exactly the same tip termination. Two molecules show different STM contrasts.

In the case of the FePc molecule located on the N-dopant, there is no evidence of symmetric steps in dI/dV spectra that would reveal the spin excitation. This observation fully supports our claims that the molecule is found in singlet state, where the spin-flip excitations are not expected.

On the other hand, in the case of FePc on the pristine graphene, we observe quite distinct scenario, revealing a step-wise increase of the conductance at symmetric bias voltage ± 7 meV. This feature can be interpreted as inelastic spin excitations of the $S = 1$ spin multiplet due to spin-orbit interaction giving rise to zero field splitting (see N. Tsukahara et al PRL 102, 167203 (2009)). We would like to emphasize, that very similar spectra were also obtained for Fe-tetraphenyl-porphyrin molecule (see J. Li et al., Sci. Adv. 2018; DOI: 10.1126/sciadv.aag0582), which was attributed to inelastic spin excitation of the $S = 1$ spin state. The presence of the triplet state of FePc on pristine graphene is also predicted by our total energy DFT calculations.

Therefore, we believe that the additional inelastic spin excitation measurements fully support the spin transition of FePc from triplet (pristine graphene) to singlet (on N-dopant) state.

Figure R1. a) STM topography image of FePc molecules on pristine graphene (blue) and on a N-dopant (red). The FePc molecule on pristine graphene is immobilized near a graphene bubble. Image parameters: $V_b = -2.0$ V; $I_t = 10$ pA. b) dI/dV spectroscopy near the Fermi level on both FePc molecules adsorbed on pristine graphene (blue arrow) and on N-dopant (red arrow) and on pristine graphene substrate (black) for comparison. Spectroscopies were acquired at the same tunnel resistance (1 M Ω) with voltage modulation of 0.5 mV at 170 Hz.

Action:

We have added the experimental measurements supporting the spin state transition and its discussion in the supplemental information of the manuscript in section G. We also included the discussion of the inelastic spin excitation spectroscopy in the manuscript.

On page 15, we added following paragraph: *“To reinforce our interpretation of the spin transition, we carried out inelastic spin excitation spectroscopy. We acquired dI/dV spectroscopy of FePc molecules*

adsorbed on pristine graphene and on a nitrogen dopant near the Fermi level. The corresponding spectra, shown in Fig. S11, were obtained with the same tip termination. Two molecules show the distinct characteristic STM contrasts, very similar to those presented in Fig. 2, indicating the different electronic configuration. The dI/dV spectroscopy (Fig. S11b) acquired for these two molecules with the same tip termination reveals very distinct character.

In the case of the FePc molecule located on the N-dopant, there is no evidence of symmetric steps associated with the spin excitation. This observation fully supports our claims that the molecule is found in singlet state, where the spin-flip excitations are not expected.

On the other hand, in the case of FePc on the pristine graphene, we observe a symmetric step-wise enhancement of the conductance at bias voltage approx. ± 7 meV. This feature can be interpreted as inelastic spin excitations of the $S = 1$ spin multiplet due to spin-orbit interaction giving rise to zero field splitting [43]. We would like to stress, that very similar spectra were also obtained for Fe-tetraphenylporphyrin molecule [54], which was attributed to inelastic spin excitation the $S = 1$ spin state. Therefore, we believe that the additional inelastic spin excitation measurements fully support the scenario of the spin transition of FePc from triplet (pristine graphene) to singlet (on N-dopant) state.”

Reviewer #2

The authors present here a combined experimental and theoretical work on control of spin state in a FePc molecule deposited on nitrogen doped graphene. In particular, they show that when the molecule is positioned on top of a nitrogen atom, it experiences a spin transition of the central Fe atom, from the triplet to the singlet state. This particular behavior has been probed using a CO decorated AFM tip, which reveals different intramolecular features with respect to the spin state of the molecule, and therefore with respect to the positioning of the molecule on top of a nitrogen atom or not. Moreover, these experimental findings are supported by DFT calculations, exhibiting the non-covalent character of the molecule-surface or molecule-nitrogen interaction.

I find this article very interesting and very innovative, since, as claimed by the author, the spin state of the molecule can be controlled by the sliding of this latter on the graphene surface using a standard STM tip. Also, the probe of the spin transition inside the molecule is performed in a very elegant way using the CO decorated AFM tip that shows the spin transition through different features inside the molecule. Consequently, I think this article deserves to be published in Nature Communication. I have however some questions or comments in order to understand well the Physics of this system as well as to improve the manuscript.

We are thankful to the Reviewer for his constructive comments. We learnt with pleasure that the Reviewer recognized the importance of our paper. The response to his/her concerns is given below.

To my opinion, the fundamental point that has to be explained in more details is the physical origin of the spin transition when positioning the molecule on top of a nitrogen atom. Indeed, in their abstract, the authors talk about “hybridization of the d-orbitals of iron and the p_z orbital of a single graphitic nitrogen” and then on page 15 when explaining the obtained result, they talk about “weak intermixing between orbitals with z-component ... of the N-dopant ... and ... Fe”.

These two assumptions seem to be rather contradictory, since in the first case one would think about a covalent bonding, whereas in the second, one would think more in an electrostatic effect. At least one of the two sentences has to be corrected. But more importantly, I would request the authors to define more this interaction. Is this due to van der Waals forces between the molecule and the nitrogen atom? Is it only an electrostatic effect at the van der Waals distance between the molecule and the nitrogen atom? How do the authors define the concept of “weak intermixing between orbitals”, and besides, what would

be a “strong intermixing”? Also, what is the interaction responsible for the promotion of an electron from the orbital d_{z^2} to the orbitals (d_{yz} , d_{xz})?

According to our DFT calculations, there is not any covalent dative bonding between the Fe atom of FePc and the N atom of doped-graphene. Such a bond is typically formed between strong ligand as, e.g., the CO molecule (than Fe-C distance is ~ 2 Å), which can cause a spin transition. In the present case, the calculated distance between the Fe atom and the N-dopant is ~ 3.7 Å. Consequently, there is only a weak interaction between Fe and the N-dopant. This claim is fully supported by the Wiberg bond analysis, which gives low Wiberg bond index ~ 0.01 (between Fe and N-dopant) compared to 0.57 for dative bonds between Fe and N atoms of phthalocyanine ring within FePc molecule. The stabilization of the complex is driven dominantly by a nonspecific dispersion (van der Waals) energy. In the case of the N-dopant, there is also additional minor contribution of the electrostatic force, but the bonding distance between FePc and graphene remains almost the same.

These results of the DFT calculations are fully supported by the experimental frequency shift vs. distance spectroscopy (see Figure S9), which reveals that FePc molecule is located approx. 3.3 Å above both the pristine graphene and N-dopant. Such distances between the FePc molecule and substrate rule out the presence of any covalent/dative bond.

We can also rule out the role of the electrostatic field based on (i) a calculation of free standing FePc in an external electric field generated by positive monopole charge located 3.7 Å (mimicking position of N-dopant) from FePc molecule, which predicts the $S=1$ triplet state as the ground state; and (ii) the case of double N-dopant in *para* position (two nitrogen atoms in *para* position, see Fig. S10), which exhibits even stronger electrostatic effects than the single N-defect, but does not show any spin transition with respect to pristine graphene.

We explained the spin transition by weak intermixing between orbitals with z-component of N-graphene (p_z orbital of N-dopant) and of FePc (d_{xz} , d_{yz} , d_{z^2}) resulting in reshuffling of Fe d-orbitals energies/occupancies. The reshuffling of the z-component d orbitals is facilitated by the fact they are close in energies.

To understand in more detail the origin of the orbital reordering driven by intermixing of p_z orbital of N-dopant and d_z -like (d_{xz} , d_{yz} , d_{z^2}) molecular orbitals of FePc, we analyzed electronic structure of the single substitutional N-dopant and its extension into the vacuum with respect to pristine graphene. The results are shown on Figure R2. In the case of the N-doped graphene we observed a quasi-localized state in the energy range of $-5.8 - -5.4$ eV. This localized resonance has dominant contribution of p_z -orbital of the N-dopant atom. What is even more relevant, we analyzed the spatial extension of the localized state of the N-dopant atom towards to vacuum with respect to the linear p_z -band of the pristine graphene, as shown on Fig. R2 a,b. Clearly, the charge density in vicinity of the N-dopant protrudes much more than this on the C atoms of the pristine graphene. This effect is also reflected in lowering of Wiberg bond index between FePc and pristine graphene by one half with respect to the case of FePc on N-defect.

Consequently, we propose that the charge density localization near the graphitic N-defect and its extension out of the surface is responsible for the spin transition. It enhances interaction between the protruding p_z orbitals of the graphitic N dopant and three adjacent C-atoms and z-component d orbitals of Fe contrary to the pristine graphene. Note that this scenario does not hold for *para* N-defect, because its localized state is located much deeper in energies (~ 1 eV), sufficiently far from the Fermi level. Therefore it is not involved in the interaction mechanism.

Figure R2.: Isosurfaces (0.003e) of calculated real space charge density of pristine graphene (a) and graphitic N-doped graphene (b) in the energy range from -5.8 to -5.4 eV (marked by grey transparent column in Fig. (c)); (c) calculated density of state (DOS) for the pristine and graphitic N defect. The dotted vertical lines denote the Fermi level for each system. The dashed line represents projected DOS corresponding to p_z -orbital of N atom. The results were obtained with 6x6 unit cell sampled with 144 k-points.

Action:

Accordingly, we have modified the respective sentences in the revised manuscript. Specifically, the sentence from the Abstract starting “*The spin transition ...*” was replaced by the following one: “*The spin transition was explained by weak intermixing between orbitals with z-component of N-graphene (p_z of N-dopant) and of FePc (d_{xz} , d_{yz} , d_z^2) resulting in the reordering of Fe d-orbitals.*”

Further, we added Figure S3 in SOM, which provides detail information about spatial distribution and DOS of pristine and N-doped graphene. In addition, on p. 16 in discussion of results obtained from DFT calculations we added “*On the other hand, analysis of electronic structure of pristine and N-doped graphene reveals, in the case of graphitic N-dopant, a presence of a localized state near the Fermi level, see Fig. S3 c). The charge density of this localized state protrudes out of surface substantially compared to the pristine graphene, see Fig. S3 a) and b). Therefore, this charge density localization enhances interaction between the protruding p_z orbitals of the graphitic N dopant and three adjacent C-atoms and z-component d orbitals of Fe contrary to the pristine graphene. This effect is also supported by lowering*

of calculated Wiberg bond index between FePc and pristine graphene by one half with respect to the case of FePc on N-defect. Consequently, we attribute the orbital reordering and the related spin transition to this effect of charge localization.”

In that respect, I am also a bit confused with the spin repartition in the d orbitals of the triplet state. Indeed, naively one would expect an extra spin down in the dz² orbital letting two spins up in the (dyz,dxz) orbitals, gaining energy. Can the authors comment about it?

In the original manuscript, we already mentioned that the triplet state characterized by the iron d-orbital occupation $d_{xy}^2 d_{z^2}^2 d_{xz}^1 d_{yz}^1 d_{x^2-y^2}^0$ provides the same scenario. Since, d_{z^2} , d_{xz} and d_{yz} are close in energy, so, $d_{xy}^2 d_{z^2}^2 d_{xz}^1 d_{yz}^1 d_{x^2-y^2}^0$ and $d_{xy}^2 d_{z^2}^1 (d_{xz} d_{yz})^3 d_{x^2-y^2}^0$ are almost iso-energetic. It is obvious that any of the above mentioned triplet state occupation would be transferred to singlet state occupation in presence of N-doping.

Action:

We have accordingly modified the manuscript specifically on p. 15 after “...same scenario.” we added the following sentence :

“Since, d_{z^2} , d_{xz} and d_{yz} orbitals are close in energy, the $d_{xy}^2 d_{z^2}^2 d_{xz}^1 d_{yz}^1 d_{x^2-y^2}^0$ and $d_{xy}^2 d_{z^2}^1 (d_{xz} d_{yz})^3 d_{x^2-y^2}^0$ states are almost iso-energetic.“

This peculiar repartition could be related to some molecular distortion, and a change in the distortion due to the interaction with nitrogen could also be responsible for the spin transition by increasing the level splitting. Therefore, I would request the authors to accurately analyze the calculated geometries of the molecule on top of nitrogen and on top of pristine graphene, in order to observe some bond lengths differences or distortions. By the way, the authors talk about a possible tilt of the FePc molecule on the N-dopant, this tilt could also be the expression of this geometry change due to the interaction with the N dopant.

Following the reviewer’s suggestion, we have analyzed the geometries of FePc on top of N-doped and pristine graphene surfaces. In both cases, FePc and the surfaces remain planar upon interaction. However, in the case of graphitic N-dopant, there is a slight shortening of Fe-N bond (0.01 Å) in FePc, which is accompanied by the spin transition. Also, the whole molecule is slightly tilted, due to asymmetric position of the N-dopant with respect to center of the molecule. However, importantly, we do not observe any vertical relaxation of Fe atom out of the molecular plane.

It should be noted, that it was predicted (Phys, Rev. Lett. 107, 257202) that by applying a reversible strain (up to 2%) to an iron porphyrin (FeP) molecule deposited at a divacancy site in a graphene lattice it is possible to switch from the high spin ($S = 2$) to the intermediate spin ($S = 1$) state. Further, the spin-transition was associated to a change in Fe-N bond length in FeP. To test this hypothesis, we constrained the magnetic moment on Fe in the gas-phase FePc, FePc on pristine graphene, and on N-doped graphene and relaxed the molecule’s geometry. Consistently, we observed an increase of the Fe-N bond length by ~1% accompanying the spin transition between spin-singlet and spin-triplet states. Further, constrained magnetic moment calculations revealed a further stretch of the Fe-N bond length by ~4% associated with the transition to the high-spin state ($S = 2$).

However, we believe that the Fe-N bond distance shortening is the consequence of the spin transition (i.e. orbital reordering), but it is not its origin.

Action:

We modified an original text on page 13: “Since our DFT geometry optimization calculations showed negligible internal relaxation of the molecule located on either pure graphene or the N-dopant, we can rule out any effect related to the central Fe atom height variations.” as follows: “*Our total energy DFT calculations showed a tiny shortening of internal Fe-N bond by ~1% when located on the N-dopant accompanying the transition between the triplet and singlet states. The whole molecule is slightly tilted, due to asymmetric position of the N-dopant with respect to center of the molecule. However, importantly, we do not observe any vertical relaxation of Fe or N atoms out of the molecular plane. Therefore, we can rule out that the contrast variation is caused by the internal vertical relaxation of Fe and N atoms out of molecular plane.*”

I think also that a representation of the calculated isoelectronic density of states between the graphitic nitrogen and the iron atom, compared to the case of standard graphene and the iron atom is very important to better understand this concept and more particularly, the non-significant overlap. Actually, it seems believable that at 3.7 angströms distance the interaction is so weak that there is no significant overlap, but on the other hand, it is more complicated to understand the important energy shift of the dz² orbital and the consequent electron transfer leading to the spin transition. Consequently, such a representation might help the reader to understand this process.

We thank the Referee for this comment and suggestion, which helped us to provide more insight into underlying mechanism of the spin transition. We already addressed this comment in above discussion of the real space charge densities of pristine and graphitic N-doped graphene.

Action:

Please see the previous response above.

I also have some minor comments. Regarding the methodology, what is the interest in performing cluster with a circumcircumcoronene molecule and periodic calculations instead of using directly the graphene unit cell and periodicity? In particular, the authors have used cluster calculations to prove that there is no significant charge transfer, but does the charge transfer between the molecule and the circumcircumcoronene reproduce well the charge transfer between the molecule and graphene?

Besides the periodic-boundary calculations we also used the cluster model in order to benefit from additional analyses implemented for cluster calculations like Natural Bond Analysis. The respective calculations helped us to analyze stabilization energies and the nature of bonding in the studied systems. It is worth noting that the possibility of inter-molecular charge transfer was discarded because of the location of the dI/dV resonances far from the Fermi level (Fig. 2b in the main manuscript) and by our Kelvin probe force microscopy (KPFM) measurements. We also used the cluster calculations to assess the energies with different spins using constrained DFT (cDFT), where we have restricted the charge transfer between the two fragments. Remarkably, there is still spin change for FePc in presence of N-dopant. Therefore, we can exclude the intermolecular charge transfer as a possible reason of spin crossover.

Finally, there are several typos in the text that need to be corrected, the legend of Figure 3 (E) has to be corrected (confusion between red and green curves), and the energy units should probably be set in eV/molecule.

In the revised manuscript, we have removed the typos and other points mentioned have also been taken care of.

Reviewer #3 (Remarks to the Author):

In the manuscript “Non-Covalent Control of Spin-State in Metal-Organic Complex by Positioning on N-Doped Graphene and Its SPM Discrimination”, the authors Bruno de la Torre, et al. study the influence of the presence of N-dopants in graphene on the adsorbed iron(II)phthalocyanine (Fe-Pc) molecule.

They use a non-contact atomic force microscope with 5 K base temperature to measure the local contact potential difference with Kelvin probe between Fe-Pc molecules adsorbed on top of a N-defect or on top of pristine graphene. Furthermore, by functionalizing the tip apex with a CO molecule they detect between molecules adsorbed on these two sites saddle differences in the frequency shift channel of constant height images. Using DFT calculations they claim that the difference is due to a change of the spin state of the magnetic molecule.

Unfortunately, the authors fail to prove their claim. From the experimental point of view I am surprised that the authors haven't performed dI/dV measurements around the Fermi-edge with higher resolution as the ones shown in figure 2b. At small voltages the spin states of Fe-Pc can be revealed either via spin-excitations as seen p.ex. in ref. 43 for Fe-Pc on Cu(110)-(2x1)-O or by the presence of the Kondo state as seen p.ex. in ref. 44 for Fe-Pc on Au(111). Such a measurement would enable to unambiguously prove their claim.

Here I would like to mention that many experiments have seen 2 or more types of Fe-Pc molecules depending on the precise adsorption configuration on different surfaces. For example Tsukahara et al. (ref. 43) found to different adsorption sites on Cu(110)-O which changed the magnetic anisotropy of the molecule but NOT the spin state. Gao et al. (PRL99,106402(2007)) was the first who observed site dependent Kondo features for Fe-Pc on Au(111). Subsequently, Ben Warner et al. (Nat.Nano 10, 259 (2015)) showed that on CuN/Cu(110) the molecule can exhibit strong negative-differential conductance with clear magnetic origin.

To summarize, only observing two different topographic appearances is not sufficient for the claim of changing the spin state. Therefore, in the present form I can not recommend the paper for publication. However, if the authors provide additional evidence then the situation might change.

Authors reply:

We really appreciated reviewer's comments, which definitively helped us to improve the manuscript. Following his/her suggestion, we performed inelastic spin excitation spectroscopies, displayed in Figure R1 and discussed above in reply to comments of the Referee #1. We hope that the distinct signal observed in inelastic spin excitation spectroscopy will convince the Referee #3 about the proposed scenario of the spin transition.

We also agree that only a topographic appearance is not sufficient to address a change in the molecular spin state. However, we would like to stress two important aspects: (i) the variation of AFM contrast is not related to a topographic (atomic structure) change of molecule but to change of electron density (reflecting reordering of occupancy of z-component d-like molecular orbitals); (ii) beside this evidence we also provided KPFM, STS, AFM and DFT measurements. All of them point to a change in the electronic structure of the molecule. However, we understand that a deeper investigation using alternative technique such as inelastic spin spectroscopy may provide an even stronger evidence of the spin state transition.

Action:

We have added the additional experimental measurements supporting the spin state transition and its discussion in the supplemental information of the manuscript in new section G. We also included the results of inelastic spin excitation spectroscopy in the discussion in the manuscript.

Referee 3 comment:

Apart from the main critique I would like to authors prior resubmission to clarify the following point.

- Figure 2a shows the manipulation of Fe-Pc. Why are the N-defects which are clearly visible in (1) no longer visible in (2)-(4)? What is the dashed line indicating?

Authors reply:

To reveal the exact position of the N-dopant on graphene, we had to reduce the bias voltage to near the Fermi energy (around 50mV). On the other hand, the optimal bias for imaging of the molecules needs to be much higher ($\sim -2V$) to ensure electron tunneling through the molecular orbitals. Therefore, we switched the bias accordingly to image either N-dopants or the molecules. The dashed line in the Fig 2a (1) denotes the transition point, where we changed the bias.

Action:

In order to clarify the situation we have added the next sentence to the main text of the manuscript: “...*the exact position of the N-dopant was resolved in situation 1 by changing the bias voltage to -0.05V below the plotted yellow dashed line.*“

Referee 3 comment:

- Figure 3 shows KPFM measurements. Why does the map in (b) has a dynamic range of $\sim 0.2V$ but the $df(V)$ measurements on the dopant and pristine graphene varies only by $\sim 9mV$? What are the errors of the parabolic fit? Is this small difference outside the uncertainties? Panel (e) also shows an STM image of two molecules on which the $df(V)$ spectra have been obtained. Why did the authors chose two molecules which clearly overlap?

Authors reply:

We agree with the reviewer that the actual plot of the KPFM measurements can induce confusion. Here we would like to replay review questions one by one to clarify the observed differences:

- 1- As KPFM maps the work function of the explored objects, this should decay with the tip-sample height. Indeed, for large tip-sample separations, the mesoscopic character of the tip averages the contribution of the different species on the surface. In Figure 3b, when we map variation of the local contact potential difference (V_{lcpd}) of a single N-dopant, the tip height was reduced in order to explore the local effect of the dopant in the graphene substrate. However, the KPFM spectra plotted in figure 3e were acquired at constant height. Because of the large adsorption height of the molecules on the graphene (about 335 pm) we separate the tip from the case displayed in Figure 3b. Despite of that, the difference in the work function between the N-dopant and the graphene is still observed.
- 2- Our errors in fitting parabolas vertexes are about 1.5 mV. So, while the small difference observed between molecules is within the uncertainty range, that is not the case for the difference between the N-dopant and the graphene at the same height.
- 3- Interestingly we did not observe any influence on the electronic structure of molecules located near a molecule on a N-dopant of those adsorbed directly on pristine graphene. Only the molecule adsorbed directly on the N-dopant shows the characteristic closing of the molecular gap observed in dI/dV spectroscopy. Indeed, Figure S7 (a) shows a very similar system where two molecules forming a dimer are displayed with only one of those molecules adsorbed directly on a N-dopant.

In this case, the AFM image of the molecule on the N-dopant shows the square-like feature (singlet state) while the near molecule shows the cross-like feature (triplet state).

Action:

With the aim to clarify points 1 and 2 of the reviewer's comment we modified Figure 3 and expand the information provided in Figure caption with the following sentences:

-“(E) Frequency shift dependence with bias voltage acquired at the center of a FePc molecule adsorbed on pristine graphene (blue) and at a N-dopant (red), as well as for a N-dopant molecule (green) and C atom in pristine graphene (black). The four LCPD measurements were acquired at the same tip-sample distance and errors in fittings are about 1.5 mV”.

-“The data plotted in (B) and (E) were acquired in different sessions with metallic tips at different tip-sample separations”.

Referee 3 comment:

- In lines 255-257 the authors write: “This is a remarkable result since it demonstrates the capability of the high-resolution AFM technique to distinguish between different molecular spin states without the need for a spin-polarized tip.” This sentence is not correct. It is not necessary to have a spin-polarized tip for distinguishing between different spin states. It is broadly used state-of-the-art to determine the spin state with dI/dV measurements.

Authors reply:

We thank the Reviewer pointing out this inaccurate claim and we agree with reviewer that not only spin-polarized method permits to distinguish the molecular spin states. As mentioned, also the state-of-the-art measurements of inelastic spin excitation at low temperatures may resolve the molecular spin state. In general, such experiment requires an advanced experimental STM setup including magnetic field, which in general enables unambiguously distinguish spin excitation signal from vibrational excitations by application of a magnetic field.

The aim of our work is to extend high resolution imaging with CO functionalized tips, which is a well-established non-contact AFM technique, to the new capability of molecular spin discrimination. This can allow simultaneous elucidation of both chemical structure and spin state of a given molecule. Here we would like to add that the use of a functionalized CO tip permits to resolve the chemical structure of an observed molecule but low energy vibrational excitations of the CO molecule may hinder a proper detection of the spin excitation of the observed molecule [see PRL **119**, 166001 (2017)].

It also worth mentioning, that possible detection of the inelastic spin excitation signal corresponding to f-electron states is still under debate and it has not been proven yet. This is attributed mainly to larger spatial localization of f-electronic states than d-electron states, which substantially suppresses the spin excitations signal. In high-resolution AFM technique, functionalized probe operates in very close tip-sample distances, where it experiences strong Pauli repulsion. Therefore, it enhances possibilities to sense differences in occupancies of f-electron states, similarly to those observed in the case FePc molecule. Thus, it would be interesting to explore if the high-resolution AFM technique would be able to resolve such spin transition including f-electron states.

Action:

We have replaced the sentence pointed out by the reviewer by the following one:

“ This is a remarkable result since it extends already outstanding capabilities of high-resolution AFM technique by a possibility to discriminate different molecular spin states without need for magnetic tips.”

REVIEWERS' COMMENTS:

Reviewer #1 (Remarks to the Author):

In the revised manuscript the authors reply comprehensively to all raised points in the previous round of review. In particular, they present new data regarding the spin state of the Fe centers by acquiring tunneling spectroscopy data in a small energy window. This clearly shows that the Fe ions present different spin states. The presentation and discussion of the results has been amended accordingly in the manuscript.

Therefore, I can recommend publication of the manuscript in Nature Communications.

I would like to add that to avoid confusion in the interpretation the authors should make it more clear that the AFM actually detects a change in electronic configuration that is (later) associated to the change in spin state. It does not directly discriminate a spin state change. There is a fine distinction between the lines of reasoning. A detection of a change in AFM signal does not automatically mean a change in spin state.

This can be implemented in the sentence in the lines 262-264.

Reviewer #2 (Remarks to the Author):

The authors have satisfactorily answered to my questions and comments and therefore I recommend the article for publication in Nature Communications.

Reviewer #3 (Remarks to the Author):

I am happy to see that the authors followed my suggestion and now provide some direct spectroscopic measurements which support their claim of a spin change between the two species. It is, however, a pity that these spectra is only shown in the supplemental material.

Therefore, I would highly recommend the authors to consider publishing this part also in the main text. Apart from a direct prove of inelastic excitations in the FePc/G spectrum, the Figure S11 contain even more interesting aspects. For example, the strong asymmetry of the steps between positive and negative bias coincide with the strong increase of density of states in their bare graphene sample.

Reviewer #1 (Remarks to the Author):

In the revised manuscript the authors reply comprehensively to all raised points in the previous round of review. In particular, they present new data regarding the spin state of the Fe centers by acquiring tunneling spectroscopy data in a small energy window. This clearly shows that the Fe ions present different spin states. The presentation and discussion of the results has been amended accordingly in the manuscript.

Therefore, I can recommend publication of the manuscript in Nature Communications.

I would like to add that to avoid confusion in the interpretation the authors should make it more clear that the AFM actually detects a change in electronic configuration that is (later) associated to the change in spin state. It does not directly discriminate a spin state change. There is a fine distinction between the lines of reasoning. A detection of a change in AFM signal does not automatically mean a change in spin state.

This can be implemented in the sentence in the lines 262-264.

Incriminated lines have following content: *“Although the case of FePc may seem very specific, we want to stress that a similar approach should be possible whenever changes in the spin state are associated with a considerable local change in electron density within the molecule.”*

We are convinced that there is nothing wrong with the content of this sentence. Similarly, any IETS peak cannot be directly associated with a spin transition, unless a careful analysis in magnetic field is carried out. In this sense, complementary high-resolution AFM measurements can be optionally used to confirm the spin transition, when measurements in magnetic field are not available.

We accomplish the referee’s request, we added a sentence.

“However, unambiguous confirmation of the spin cross over should be always confirmed by independent measurements, such as inelastic spin excitation spectroscopy.”

Reviewer #3 (Remarks to the Author):

I am happy to see that the authors followed my suggestion and now provide some direct spectroscopic measurements which support their claim of a spin change between the two species. It is, however, a pity that these spectra is only shown in the supplemental material.

Therefore, I would highly recommend the authors to consider publishing this part also in the main text. Apart from a direct prove of inelastic excitations in the FePc/G spectrum, the Figure S11 contain even more interesting aspects. For example, the strong asymmetry of the steps between positive and negative bias coincide with the strong increase of density of states in their bare graphene sample.

Following the referee suggestion, we moved the Figure S11 into the main text now figure 5.